Expanding walls and shrinking beaches: loss of natural coastline in Okinawa Island, Japan

Masucci Giovanni Diego giovannimasucci@me.com 1
Reimer James D. 1 2
1 Molecular Invertebrate Systematics and Ecology Laboratory, Graduate School of Engineering and Science, University of the Ryukyus , Nishihara , Okinawa , Japan
2 Tropical Biosphere Research Center, University of the Ryukyus , Nishihara , Okinawa , Japan
Pochon Xavier
Electronic publication date: 2019 Sep 6
Publication date: 2019
Volume: 7
Electronic Location ID: e7520
Received 2019 May 30; Accepted 2019 Jul 19
Copyright: ©2019 Masucci and Reimer
Copyright year: 2019
Copyright holder: Masucci and Reimer
License: This is an open access article distributed under the terms of the Creative Commons Attribution License, which permits unrestricted use, distribution, reproduction and adaptation in any medium and for any purpose provided that it is properly attributed. For attribution, the original author(s), title, publication source (PeerJ) and either DOI or URL of the article must be cited.
License URL: https://creativecommons.org/licenses/by/4.0/

Keywords: Habitat loss, Habitat fragmentation, Land reclamation, Coastal armoring, Coral reefs, Okinawa Island, Japan, Land-filling, Anthropic impact, Remote sensing

Funding: The authors received no funding for this work.

==============================
Okinawa is the largest and most populated island of the Ryukyu Archipelago in southern Japan and is renowned for its natural resources and beauty. Similar as to what has been happening in the rest of the country, Okinawa Island has been affected by an increasing amount of development and construction work. The trend has been particularly acute after reversion to Japanese sovereignty in 1972, following 27 years of post-war American administration. A coastline once characterized by extended sandy beaches surrounded by coral reefs now includes tracts delimited by seawalls, revetments, and other human-made hardening structures. Additionally, part of coastal Okinawa Island was obtained by land-filling shallow ocean areas (land reclamation). Nevertheless, the current extension of the artificial coastline, as well as the level of fragmentation of the natural coastline are unclear, due to the lack of both published studies and easily accessible and updated datasets. The aims of this research were to quantify the extension of coastline alterations in Okinawa Island, including the amount of land-filling performed over the last 41 years, and to describe the coastlines that have been altered the most as well as those that are still relatively pristine. The analyses were performed using a reference map of Okinawa Island based on GIS vector data extracted from the OpenStreetMap (OSM) coastline dataset (average node distance for Okinawa Island = 24 m), in addition to satellite and aerial photography from multiple providers. We measured 431.8 km of altered coastline, equal to about 63% of the total length of coastline in Okinawa Island. Habitat fragmentation is also an issue as the remaining natural coastline was broken into 239 distinct tracts (mean length = 1.05 km). Finally, 21.03 km2 of the island’s surface were of land reclaimed over the last 41 years. The west coast has been altered the most, while the east coast is in relatively more natural conditions, particularly the northern part, which has the largest amount of uninterrupted natural coastline. Given the importance of the ecosystem services that coastal and marine ecosystems provide to local populations of subtropical islands, including significant economic income from tourism, conservation of remaining natural coastlines should be given high priority.

Introduction

Japan is renowned worldwide both for its economic, industrial, and technological achievements, and for its cultural and natural heritage. Unfortunately, a consequence of the Japanese development model has been the destruction of portions of the country’s natural environments, both terrestrial and marine (McCormack, 1999; Kerr, 2001). After World War II, Japan has increasingly invested public money in projects related to modifications of the natural landscape (Walker & Mossa, 1986). This included the alteration of 110 out of its 113 major rivers with numerous dams and concrete enclosures, in addition to minor streams across the country, and the creation of numerous mountain roads for the forestry industry, which has contributed to the eradication of around 40% of Japan’s native forest (Kerr, 2001). During the 1990s, the production of concrete in Japan rose, reaching the highest cement consumption per capita in the world and the highest per capita construction-related debt (¥8 million; Kingston, 2005). This increase was seemingly unaffected by the recession that the country was facing, which, instead, accelerated the production: between 1992 and 1999, the country invested ¥120 trillion to boost the economy, of which ∼60% was spent on construction projects (Kingston, 2005), producing as much concrete per year as the United States (Kerr, 2001; Kingston, 2005). During this time, local and international commenters began describing Japan as the ‘doken kokka’, the construction state (McCormack, 1999; Kerr, 2001; Kingston, 2005; Nam, 2019).

Perhaps the most well-known and evident impact of construction can be seen on the coastline, where numerous civil engineering projects have been performed, either by the placement of bulkheads, seawalls, breakwaters, revetments, and groins, with the purpose of protecting human-made buildings and coastal roads from erosion and wave action (shoreline hardening, or armoring), or by extending the coastline seaward by filling the ocean with soil, rocks, cement, or combinations of these materials, in a process known as “land reclamation” or “land-filling”. Both of these expressions are not without issues: the term “land reclamation” has also been associated with a wide range of human-made alterations, generally with the aim of converting disturbed land to productive uses (Powter, 2002). In a similar way, the term “land-filling” overlaps with the practice of disposing garbage by burial on land. In the context of this research, the meanings of the two expressions are identical, but we here use the more neutral and descriptive term “land-filling”.

When a coastline is eroding, coastal armoring can interfere with natural beach upslope migration, triggering a passive erosion of the shoreline that can lead to beach narrowing or loss (Griggs, Tait & Corona, 1994; Dugan et al., 2008), a phenomenon which has been defined as “coastal squeeze” (Pat, 2004). In terms of impacts on the marine ecosystems, shoreline hardening reduces habitat complexity and uniqueness, affecting species composition, and decreasing abundance and diversity (Seitz et al., 2006; Dugan et al., 2008; Gittman et al., 2016; Aguilera, 2017). The effects of land-filling the marine environment include habitat loss (Lai et al., 2015; Heery et al., 2018) and a degradation of environmental parameters levels, particularly higher nutrients, sedimentation rates, and turbidity, which can negatively impact the survival of coral reefs (Chou, Yu & Loh, 2004; Dikou & Van Woesik, 2006).

Japan’s “Environmental Impact Assessment (EIA) Law” was enacted in 1997 and implemented in 1999, providing legal power to deny authorization to construction projects exerting excessive impacts on the environment (Ministry of the Environment of Japan, 2019). Although an important achievement, the law has some limitations that need to be considered. Only large-scale construction projects require EIAs. For medium-scale works, the necessity of an EIA is judged by the national government for each individual project, while smaller-scale projects do not require any assessment by national law. In the case of land-filling, an EIA is only required for projects involving over 50 ha (0.5 km2) of reclamation. For land-fills between 40 and 50 ha (0.4–0.5 km2), the decision is taken on a per-project basis, and below 40 ha (0.4 km2) there is no requirement. If the reclamation is performed to build harbors and ports, the limit for no EIA is extended to 300 ha (3 km2). Although the size of a project is an important factor, it is not the only one: EIA only regulates specific categories of construction works. While land-filling is one of the covered categories, coastal armoring is not included and can therefore proceed without any EIA (see Ministry of the Environment of Japan (2019) for a detailed list of categories and sizes subject to EIA Law).

EIAs are performed by the project proponent and sent to an authorizing agency (commonly the Ministry of Infrastructure, Land and Transport, or the Ministry of Economy, Trade and Industry), which makes the final decision. Because the authorizing agency and the proponents can share similar interests, the Ministry of the Environment (MoE) can be requested to express an opinion. However, a positive MoE opinion is not required for the final approval.

Finally, there is no requirement for follow-up surveys that would provide data on the quality of the initial EIA and on the impacts of the construction work on the natural environment (Ministry of the Environment of Japan, 2019). This is one of the reasons why the impacts on the natural environment of coastal construction projects in Japan are generally unreported, and thus unclear or unknown. EIAs are not the only legislative instruments that regulate construction works; in the case of land-filling, a permit from the local government is needed, making the role of prefectural governors important in the final decision (Okinawa Prefectural Government, 2014; Ministry of Land, Infrastructure and Transport of Japan, 2001).

The Ryukyus (also Nansei Islands), a subtropical archipelago located in southwestern Japan, consist of approximately 160 islands. They include the most extensive coral reefs of the country, hosting almost 90% of the scleractinian coral species found in Japan (∼360 species out of ∼415; Nishihira, 2004), and have high levels of marine diversity and endemism (Cowman et al., 2017). Okinawa Island, or Okinawa-jima in Japanese (Fig. 1), is the largest and most populated island in the archipelago (area = 1,208 km2; Japan Statistics Bureau, 2014; population = 1.3M people; Okinawa Prefectural Government, 2019a).

Figure 1 Okinawa Island and surrounding areas in southern Japan, northwestern Pacific Ocean.

The map includes locations discussed in this research.

The beauty of Okinawa’s natural sandy beaches, rocky cliffs, and marine environment has provided significant economic income from tourism, and there has been a rapid growth in the number of visitors in recent years, surpassing those of Hawaii in 2017 (total number = 9,579,900 visitors; Ryukyu Shimpo, 2018). However, the natural environment has also been damaged by human activities, including deforestation, urban sprawl, coastal development, coastal hardening, and land-filling (McCormack, 1999; Nakano, 2004; Reimer et al., 2015; Heery et al., 2018).

Since 1972, the year Okinawa Prefecture reverted to Japanese control, construction projects have profoundly modified and reshaped the island landscape and topography (McCormack, 1998; McCormack, 1999; Kerr, 2001). By 1992, more than 1,600 ha of reef had been destroyed due to land-filling and dredging, roughly equal to the 6% of the total coral cover around the island (Nakano, 2004). The high influx of tourists has contributed to the building of new infrastructure along the coastline and on reclaimed land (McCormack, 1998; Tada, 2015). Despite the difficulties in estimating general trends and local impacts due to the lack of post-work assessments and published studies, construction has been pursued to such an extent that civil engineering has been described as the main cause of coral reef destruction in the region (Nakano, 2004). Today, the development in Okinawa Island continues and, as of January 2019, three major land-filling projects are underway: the construction of a second runway at Naha Airport (1.6 km2 of reclaimed land; Flyteam Japan, 2013), the creation of a harbor, residential facilities, resorts, and artificial beaches at the Awase Tidal Flats (2.66 km2; Nakano, 2004), and the building of a new US military base at Cape Henoko/Oura Bay (1.6 km2; Okinawa Defense Bureau, 2012).

In this research, using GIS software and remote sensing technologies, specifically satellite imagery and aerial photography, we described the current state of the Okinawa Island coastline, with three main objectives:

(1) To quantify the amount of coastline that has been altered as of the end of 2018, and to categorize such alterations. This included a comparison between the east and west coasts, as we hypothesized higher levels of impact and habitat fragmentation on the west side of the island, due to the presence of the capital and largest city Naha and of National Route 58, a coastal road constituting the main connection between the north and south of Okinawa Island.

(2) To generate a map allowing easy visualization of the current status of the island coastline, including habitat fragmentation, and to identify which parts of the island have lost the largest amounts of natural coastline, and which locations have the highest amount of preserved coastal environment.

(3) To determine the amount (area) and locations of land-filling performed during the last 41 years of Japanese administration of Okinawa Island (1977–2018).

Both the maps generated in this study and the underlying dataset provide a baseline for future studies and can be used as an additional tool when performing evaluations on projects potentially impacting the island’s natural coastal environments.

Materials & Methods

According to a survey conducted by the Japanese Coast Guard in 1986 (Japan Statistics Bureau, 2014), Okinawa Island has 476 km of coastline. However, there are two issues with this number. Firstly, the extension of a coastline is not fixed in time and can be affected by natural factors, like erosion or, at shorter time scales, by human activities, including coastal hardening and land-filling. Secondly, the extension of a coastline depends on the scale at which the measurements are done. The more detailed the scale of a map, the longer the measured coastline will be, tending towards infinite (coastline paradox; Richardson, 1961; Mandelbrot, 1967), with no clear-cut gap between what is useful and unrequired detail. Because coastlines are fractals with properties of self-similarity, it is not possible to state the length of a coastline without referring to a specific reference map or to a scale and accuracy of unit of measurement. In other words, numbers referring to the length of a coastline that do not report this additional information are of little use (Mandelbrot, 1967).

For these reasons, to assess the amount of altered coastline in a way that will make possible for future studies to make comparisons at different times or with other locations, our measures were performed over a base layer of LineString data (polylines vectors connecting georeferenced points) extracted from the OpenStreetMap (OSM) coastline dataset (Haklay & Weber, 2008) at maximum detail (visualized at a scale of 1:1,000, WGS-84/long datum, dataset acquired on January 26, 2019). OpenStreetMap data provide key advantages compared to datasets from numerous other sources: (1) They are easily accessible and can be freely used for research purposes, (2) they are constantly updated, allowing us to conduct repeated surveys over the years to monitor trends and changes in the coastlines, and (3) they are not specific to one country or region, which means that they can be used for future comparisons with other islands or coastlines.

The OpenStreetMap coastline dataset (tag: natural=coastline) includes worldwide vector data delineating the sea edge, which is marked by the mean high water spring line. Coastline information is acquired from satellite data (such as NASA Landsat; see https://wiki.openstreetmap.org/wiki/Potential_Datasources#Shoreline_databases for a comprehensive list) and automatically converted into vectors using automatic image recognition algorithms. The acquired vector lines are then quality-checked and refined by the members of the OSM community. Being a vector dataset, scale (or spatial resolution) is represented by the average distance between nodes (GFC, Kansas State University, 2019). In the OSM coastline dataset, average node distance varies with location on a scale that goes from tens to a few hundreds of meters. Node density is particularly high in Europe and Japan, and lower where baseline data are more inaccurate and local contributors are fewer, such as Antarctica. Overall, the average global spatial resolution of the OMS dataset has been measured at 66 m (Hormann, 2013). Coastal generalization algorithms may be used when comparing regions of the world with different spatial resolution.

The base layer of the Okinawa coastline was imported into QGIS (version 3.6.0-Noosa; QGIS Geographic Information System, 2019) and updated with minimal modifications to take into account new and ongoing land-filling works not yet indexed by the OSM project, including Naha Airport, Awase, and Oura Bay/Henoko. The total coastline included Okinawa Island and artificial land-fills connected to the main Island. Naturally occurring islands connected via bridges to Okinawa Island were not included. However, natural islands that became merged via land-fills were included in the analyses. The total coastline was then divided into west and east coasts using the northernmost (GPS = 26.875525, 128.257702, Cape Hedo) and southernmost (GPS = 26.074467, 127.676570, Cape Arasaki) points of the island, to allow comparisons between the two. Altered portions of the Okinawan coastline were tagged over the base map layer with the help of satellite data and aerial photography from ESRI, Google, Bing, and the Okinawa Prefectural Government historical GIS dataset. Where image data were unclear, observations from the 5th basic survey on conservation of the natural environment from the Biodiversity Center of Japan, Ministry of the Environment (1998) were referenced. Finally, visits to locations to confirm information were conducted as needed. Collectively, these sources allowed us to draw a map reflecting the situation of Okinawa Island at the end of year 2018. The altered coastline was obtained by tracing LineStrings over the base layer and tagging them accordingly within defined ‘alteration’ categories. This process was done using the GIS software Map Plus for iOS (version 2.8.5, Duwei Technology, 2019).

Each coastline LineString was included into one the following alteration categories (Fig. 2):

Figure 2 Coastline categories.

(A) Natural (east Kunigami). Vegetation acts as buffer between shoreline and road. (B) Soft armoring (Odo). Beach and vegetation preserved but disconnected due to the presence of human-made structures above the intertidal zone. (C) Hard armoring (west Kunigami). Roads or buildings built next to the coastline. Presence of seawalls and/or breakwaters. (D) Land-filling (Agarihama/Agarizaki). Shallow waters turned into land to increase the space available for human activities. The yellow line indicates the original shoreline before reclamation. For each category, color choices are consistent with those used in Fig. 3. Map data ©2019 Google.

1. Natural. The shore (beach or rocky shore) and the terrestrial area immediately behind the shore have been preserved, allowing the existence of a buffer zone made by vegetation and/or dunes.

2. Soft armoring. The shore has been hardened by walls or other human-made constructions. Hardening blocks are on land and the components of the natural coastline (vegetation, sand, intertidal zone, etc.) are still preserved.

3. Hard armoring. The shore has been hardened by seawalls, breakwaters, or other human-made constructions, placed into the water and/or at the interface between water and land. One or more components of the former natural coastline have been compromised by something human-made (roads, seawalls, coastal buildings, breakwaters, etc.), so that simply removing the hardening would not restore the natural coastline.

4. Land-filling. A tract of hardened coastline obtained from human-made land reclamation of the intertidal and, in some cases, subtidal zones.

It is important to note that both the map of human-made alterations and the underlying data are conservative, mainly due to two reasons. First of all, it is possible, especially in the case of soft armoring, for vegetation to grow around barriers, making them hard to confirm from satellite images. Secondly, blocks placed in the subtidal zone and completely submerged, although used in Okinawa Prefecture, could not be accounted for in our analyses as they were not reliably visible in satellite images.

LineStrings were grouped in their respective categories and imported into QGIS, where their length in kilometers was summed to obtain the total lengths (km) of each group and the percentages of each category on the total coastline. Moreover, in order to examine coastline fragmentation and the relative contributions to fragmentation (assuming all coastline was originally natural) from each coastal category above, the number of LineStrings and LineStrings mean length were calculated for the natural coastline, for the whole altered coastline, and within each category of alteration.

Descriptive statistics were performed using R software (version 3.5.3; R Development Core Team, 2019). As the dataset included the totality of LineStrings composing the Okinawa coastline, no statistical test was performed to compare west and east coast LineString mean lengths, as a statistical test would only make sense for sample data, as opposed to population parameters.

Finally, to assess the amount of land-filling performed in Okinawa Island in the last 41 years, historical aerial photography data from 1977 were acquired from the Okinawa Prefectural Government GIS webpage (Okinawa Prefectural Government, 2019b) and georeferenced. The tracts of coastline that differed from year 2018 were manually traced as vectors using QGIS. The total amount of land added by reclamation activities was obtained by subtracting the area of Okinawa Island in 1977 from that of Okinawa Island in 2018. Maps figures were generated using the QGIS PDF vector export function.

Results

The length of the coastline layer imported from the OSM dataset was 656.9 km, with an average node distance (spatial resolution) of 24 m. After adding the most recent coastal developments, the total coastline reached 682.8 km, at the same spatial resolution. 431.8 km of coastline were altered, equal to 63.2% of the total length, leaving 251.0 km in a natural state (36.8%). The most common category of coastal alteration, in terms of length in km, was land-filling (309.2 km, 45.3%), followed by hard armoring (98.9 km, 14.5%), and then soft armoring (23.7 km, 3.4%) (Fig. 3).

Figure 3 Map of human-made alterations to the Okinawa Island coastline.

Different alteration categories are represented by different colors and summarized in the pie chart, which shows their relative abundances (%). Base layer map data © OpenStreetMap contributors.

The west coast (347.5 km) was the most affected by coastal development: 251.4 km were altered (72.3%), of which 186.3 km by land-filling (53.6%), 52.8 km by hard armoring (15.2%), and 12.2 km by soft armoring (3.5%). 96.1 km were in a natural state, equal to 27.7% of the total length of the west coast. The east coast (335.3 km) was relatively more preserved: 180.5 km were altered (53.8%), of which 122.9 km (36.6%) were altered by land-filling, 46.1 km (13.7%) by hard armoring, and 11.5 (3.4%) by soft armoring. 154.8 km (46.2%) were still in a natural state (Fig. 4).

Figure 4 Coastal development categories divided between the east and west coasts of Okinawa Island.

Regarding habitat fragmentation, the number of LineStrings associated with coastal alterations was 427, meaning there were overall 427 distinct sectors of the coastline, of variable length, presenting human-made alterations and contributing to the fragmentation of the natural coast. Of these, 229 were on the west coast and 198 on the east coast. Interestingly, while land-filled LineStrings had the largest mean length (169 LineStrings, mean length = 1.83 km, sd = 4.27 km), armored tracts with no land-filling, although shorter, were more numerous (258 LineStrings, mean length = 0.48 km, sd = 0.54 km; see Table 1 for split data between soft and hard armoring).

Table 1 Summary of coastal alteration and fragmentation data.

		Natural	Soft armoring	Hard armoring	Land- filling	Total altered	
West coast	Length (km)	96.1	12.2	52.8	186.3	251.4	
% of West coast	27.7	3.5	15.2	53.6	72.3	
LineStrings number	129	32	110	87	229	
LineStrings mean length (km)	0.75	0.38	0.48	2.14	1.10	
Standard deviation (km)	1.08	0.29	0.62	5.66	3.60	
East coast	Length (km)	154.8	11.5	46.1	122.9	180.5	
% of East coast	46.2	3.4	13.7	36.6	53.8	
LineStrings number	109	31	85	82	198	
LineStrings mean length (km)	1.42	0.37	0.54	1.50	0.91	
Standard deviation (km)	2.31	0.31	0.55	1.90	1.37	
Total coastline	Length (km)	251.0	23.7	98.9	309.2	431.8	
% of Total coastline	36.8	3.5	14.5	45.3	63.2	
LineStrings number	238	63	195	169	427	
LineStrings mean length (km)	1.05	0.38	0.51	1.83	1.01	
Standard deviation (km)	1.78	0.30	0.59	4.27	2.79	

The natural coastline was composed of 238 LineStrings. The mean length of natural LineStrings was 1.05 km (sd = 1.78 km). In other words, on average, a tract of natural coastline in Okinawa Island was found to be interrupted by coastal armoring or land-filling every 1.05 km. The natural coastal environment was more fragmented in the west coast, where 96.1 km were composed of 129 distinct LineStrings (mean length = 0.75 km, sd = 1.08 km). The east coast had instead lower levels of fragmentation: 109 LineStrings for 154.8 km of natural coastline (mean length = 1.42 km, sd = 2.31 km). Overall, in the west coast, the natural coastline was composed of more LineStrings of shorter mean length, and therefore was more affected by habitat fragmentation (Fig. 5 and, for a summary of coastal alteration and fragmentation data, Table 1).

Figure 5 Natural LineStrings in Okinawa Island and their length (km).

(A) West coast. (B) East coast. The horizontal dashed line highlights the presence of a single LineString above five km on the west coast, compared to 9 LineStrings for the east coast.

The longest tract of uninterrupted natural coastline (10.71 km) was measured in the northeast area of the island, in the Kunigami District (GPS = 26.844964, 128.296063–26.794411, 128.317441). The same area hosted two additional tracts of similar length (∼10 km). The northeast of the island, from Teima, Oura Bay (GPS = 26.552018, 128.065038), to Cape Hedo (GPS = 26.875525, 128.257702), was overall the most preserved, with 82.7% of the coastline in a natural state (84.96/102.70 km). This same area also hosted the majority of natural coastline tracts above five km (8 out of 10 totals, 9 of which were on the east coast; Figure 5). For the west coast, the longest tract of uninterrupted natural coastline (6.67 km) was in the Nakijin Area (GPS = 26.703180, 127.934848–26.696041, 127.971799) with no other tracts above five km in length.

The area of Okinawa Island at the end of year 2018 was measured at 1213.35 km2, 5.35 square kilometers more than reported in the Japan Statistics Bureau (2014). This number includes land-fills detached from the main coastline (from ∼100 m to ∼1,000 m distance from shore), and connected to it via multiple bridges: on the west coast the Toyosaki land-fill (Tomigusuku; 1.41 km2; GPS = 26.156254, 127.654746), on the east coast, from north to south, the Suzaki land-fill (3.47 km2; GPS = 26.333432, 127.855115), the Awase land-fill (measured at 0.93 km2, reclamation works still ongoing; GPS = 26.303966, 127.840715), and the Agarihama/Agarizaki land-fill (1.14 km2; GPS = 26.207607, 127.765247). In 1977, the measured area was 1,192.32 km2. Hence, the island’s expansion, accountable to the land-filling projects performed over a period of 41 years, was measured at 21.03 km2, the majority of which have occurred in the southern part of the island on both the east and west coasts: 18.83 km2, or ∼90% of the land-filled area was located south of the Tancha, Onna Village and Yaka, Kin Town districts (Fig. 6). On the west coast, the area surrounding Naha, the prefectural capital has been particularly affected by coastal land-filling: from Itoman (GPS = 26.114724, 127.664448) to the Kadena US military base (GPS = 26.339156, 127.746935) the entire coastline (112.2 km) was altered, and 107 km of its length were categorized as land-filled (∼95%).

Figure 6 Map of the land-filling that occurred in Okinawa Island over a period of 41 years (1977–2018).

(A) South part of Okinawa Island (south of Tancha and Yaka). Base layer map data (2018) © OpenStreetMap contributors. (B) North part of Okinawa Island (north of Tancha and Yaka). The area from 1977 is colored in grey, newer expansions from 2018 are highlighted in red. Base layer map data (2018) © OpenStreetMap contributors.

Discussion

The results of this study revealed that over 63% of the original Okinawan coastline has been lost or altered by numerous coastal engineering projects, and that what remains is now fragmented into numerous segments divided by land-filled and armored tracts.

Land-filling, or land reclamation, was found to be the main contributor to the amount of coastline alterations (45% of the total coastline), especially in the highly urbanized south of the island, where the capital city Naha and most of the population are located. Reclamation is a direct cause of habitat loss, in particular for coral reefs and mangrove meadows. Although habitat loss as a problem is not unique to Okinawa, the situation in Okinawa Island is noteworthy, for at least two reasons: between 1993 and 1998, the largest increase (47.99%) in the amount of artificial coastline in Japan occurred in Okinawa Prefecture (Biodiversity Center of Japan, Ministry of the Environment, 1998). Secondly, unlike mainland Japan, Okinawa Island is subtropical, characterized by the presence of a well-developed fringing reef along its coastline (Reimer et al., 2019). Construction has mainly happened in the shallow waters of the inner reef, over lagoons and reef flats. Being located in the vicinity of river mouths or in the inner reef, mangrove forests and seagrass meadows have also been targets of development. In a coral reef environment, reef formations themselves are an important element in the protection of the coastline: according to Ferrario et al. (2014), coral reefs can provide significant protection from storms, dissipating on average 97% of wave energy. The economic benefit of the coastal protection provided by Japanese coral reefs has been estimated to be of 172 million USD/year (Cesar, Burke & Pet-Soede, 2003), but by reclaiming shallow reefs, this ecosystem service is inevitably lost, leaving the coastline more vulnerable and in need of artificial protection.

Similarly, seagrass meadows and mangrove forests are known to provide important ecosystem services: both, particularly seagrass meadows, act as nurseries for valuable fish species (Whitfield, 2016 provides a review on the topic). Moreover, mangrove forests offer valuable protection from coastal disasters by significantly reducing economic and human life losses during storms and tsunamis (Das & Vincent, 2009). Kathiresan & Rajendran (2005) suggested the adoption of a dense buffer zone made of mangroves and other coastal vegetation of at least one km between the ocean and the first human settlements at Parangipettai, Tamil Nadu, India. However, in Okinawa Island, numerous roads and buildings have been built just a few meters from the coastline. Several coastal roads have suffered the effects of erosion or damage due to their proximity to the ocean. In a few cases, access had to be restricted, or expensive repairs and improvements have been required for the road to remain functional, usually in the form of additional armoring (Nakano, 2004). In the northwestern part of the island, despite this being a rural area of low population density (0–100 residents/km2; Okinawa Prefectural Government, 2019b) with abundant space available to build roads, National Route 58 runs for kilometers just a few meters from the coastline (Fig. 2C). This route was built so close to the ocean that it required the employment of hard armoring, usually in the forms of seawalls and tetrapods, tetrahedral blocks of various sizes made of concrete placed in front of seawalls or piled up to form breakwaters (Hesse, 2007). This combination of coastal roads coupled with seawalls and concrete blocks have altered several tracts of natural coastline in Okinawa Island. In the case of National Route 58 and also in the north Kunigami Area, the artificial landscape extends, with only sporadic interruptions, from the northern part of the Motobu Peninsula (GPS: 26.630022, 128.029791) until Ginama Fishing Port (GPS: 26.848042, 128.253570), after which the route deviates from the shoreline. In the same northern region, but on the east coast, National Route 70 connects the north and the south of the island. Because National Route 70 was built inland, leaving a buffer space of vegetation between the road itself and the shoreline, the natural profile of the original coastline has been largely preserved (Fig. 2A). This is an important example of how different engineering approaches to similar problems can affect the environment in different ways. The different state of the east and west coasts of the north Kunigami Area shows that it is possible to build roads without sacrificing the natural coastline. In a subtropical island where tourism is the most important economic asset, and the local government aspires to become a World Natural Heritage Site (Okinawa Prefectural Government, 2019c), there should be a strong government interest in limiting the loss of beaches and natural scenery.

Although coastal armoring exerts impacts on the natural environment (Seitz et al., 2006; Sane et al., 2007; Dugan et al., 2008; Dethier, Toft & Shipman, 2017; Gittman et al., 2016; Aguilera, 2017), beyond habitat loss and fragmentation it is unclear how it can affect the general health of a coral reef ecosystem, or how communities inhabiting the surroundings of coastal armoring compare with those inhabiting natural areas. If artificial barriers actively harm corals and the wider coral reef community, then armoring a coastline would mean, in the long term, an increase in artificial defense at the expense of the protection already naturally provided by the reef. Therefore, future research should investigate in detail how shoreline armoring affects coastal ecosystems. Japan is an ideal candidate for such study, as tetrapods have been widely deployed across different prefectures (Kerr, 2001), and yet very little international peer-reviewed studies on their effects on marine communities have been published. It is clear that more research is needed to fill gaps in our knowledge, on this issue and with other coral reef conservation concerns in the region (Reimer et al., 2019).

The overall benefits of the Japanese coral reefs have been estimated as equivalent to 1665 million USD/year (Cesar, Burke & Pet-Soede, 2003). Realizing the importance of a healthy reef ecosystem, since 2012, the Okinawa Prefectural Government has been investing significant resources in coral reef restoration (6.25 million USD for the first three years of implementation; Okubo & Onuma, 2015). However, reef restoration still provides benefits estimated to be six orders of magnitude lower than the amount of damage occurring (Okubo & Onuma, 2015), and restoration results are often uncertain and, in some cases, counterproductive (Casey, Connolly & Ainsworth, 2015). For these reasons, as progress is made in restoration research and public money is invested on the effort, it is important to also better integrate conservation concerns into development plans in order to spare reefs from further destruction. Finally, it is noteworthy, in this context, that the national action plan for the conservation of Japanese coral reef ecosystems (Ministry of the Environment of Japan, 2010) mentions coastal development as a cause of habitat loss and coral reef mortality, and proposes the establishment of Marine Protected Areas (MPAs) and national parks in coral reef regions as a solution. However, as of June 2019, no Marine Protected Area exists in Okinawa Island.

Conclusions

The coast of Okinawa Island has been subject to significant alterations leading to habitat loss (63.2% of the coastline artificially altered) and fragmentation (remaining coastline divided in 239 distinct tracts of mean length = 1.05 km). As hypothesized, the west coast has been the most impacted by human development.

It is our opinion that the northeast Kunigami coast of the island should be considered as an MPA candidate, being located in an area of low human population, with a relatively pristine coastline, both in terms of amount and fragmentation, and with a biodiversity that is still largely understudied (Reimer et al., 2019).

In the future, coastal restoration/rehabilitation initiatives will need to be evaluated. We live in an era of unprecedented attention to environmental issues. In Singapore, a country where less than 20% of the coastline remains natural, mangroves have been planted with the intent of rehabilitating artificial habitats (Lai et al., 2015) and research has been made to create more variable armored intertidal zones (Loke et al., 2014; Loke et al., 2015). Several states in the US, such as North and South Carolina, have restricted or banned the use of additional coastal armoring and are planning or have performed block removals (Kerr, 2001; Miller et al., 2012; Dethier, Toft & Shipman, 2017). The “soft armoring” category of our map includes several locations where restoration activities could be feasible. In some instances (Fig. 2C) such works would be limited to the removal of old walls no longer used, reducing habitat fragmentation and restoring connectivity between land and ocean. Such restoration activities would benefit species that depend on ocean-land connectivity and have been impacted by habitat loss and fragmentation, like sea turtles for their nesting (Rizkalla & Savage, 2010), and coconut crabs to release fertilized eggs (Sato & Yoseda, 2013). Walls and barriers built years ago to protect buildings that are now abandoned, or roads no longer used and overgrown by vegetation could also be included in similar initiatives. It is anticipated this work will focus attention on the current status of the Okinawa Island coastline and be utilized as a tool for further evaluations and monitoring, tracking changes in time and allowing comparisons with other regions, in Japan or abroad.

Supplemental Information

Dataset S1 GIS Shape files (vector LineStrings) analyzed for this research

Total coastline (2018), alteration categories (2018), and Okinawa Island area at different years (1977 and 2018). Files can be opened with QGIS or other compatible GIS software.

Click here for additional data file.

The authors would like to thank Masaru Mizuyama for his suggestions that greatly contributed to the improvement of this work, Zax Zeng from Duwei Technology for providing help on acquiring and geotagging historical aerial photography data, Kohei Hamamoto for his help in translating parts of the cited Japanese literature, and Piera Biondi for her comments on early versions of the manuscript. We would like to thank the OpenStreetMap, QGIS and R software communities, whose hard work made this research possible. Comments and suggestions from Dr. Mariko Abe and two anonymous reviewers greatly improved an earlier version of this work.

Additional Information and Declarations

Competing Interests

Author Contributions

Data Availability

James D. Reimer is an Academic and Section Editor for PeerJ.

Giovanni Diego Masucci and James D. Reimer conceived and designed the experiments, performed the experiments, analyzed the data, contributed reagents/materials/analysis tools, prepared figures and/or tables, authored or reviewed drafts of the paper, approved the final draft, final English check.

The following information was supplied regarding data availability:

The raw data (GIS vector shape files) are available in Dataset S1. Dataset S1 includes LineStrings divided per category, and Polygons representing the whole Area of Okinawa Island in 2018 and 1977.

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
