# Peer review of "Expanding walls and shrinking beaches: loss of natural coastline in Okinawa Island, Japan"

_PeerJ, doi:10.7717/peerj.7520_

## Round 0.1 · original submission · Minor Revisions

Dear Giovanni and James,

I have received three positive reviews of your study. All have recognised the value and overall quality of this work, well done! However, some more detail is required on how the coastline was measured and it appears that some extra proofreading of the manuscript is required to correct some awkward English structuring.

I will be looking forward to receiving your revised manuscript, along with a point-by-point response to the reviewers comments.

With warm regards,
Xavier

Reviewer 1 ·

Basic reporting

Clear and unambiguous, professional English used throughout.
Literature references, sufficient field background/context provided.

Self-contained with relevant results.

Experimental design

Methods described with sufficient detail & information to replicate.

Research question well defined, relevant & meaningful. It is stated how research fills an identified knowledge gap

Validity of the findings

All underlying data have been provided; they are robust, statistically sound.

Additional comments

The MS of Masuccin and Reimer quantify the extension of the coast alteration in Okinawa island, considering an spatial analysis based on maps, of man made structures constructed from 51 years ago. The analyses performed showed important artificialization of the coast , with only few remant "patches" of natural habitats remaining. I found the study is very useful and well executed, and help to provide important mitigation measures to restore/manage coastal habitats of this and other scenarios.

I have no general concerns about this study, bu likely condensing results (L: 245-271) in an additional summary table (reporting natural/artificial tracts) would be easy to follow for readers.

Reviewer 2 ·

Basic reporting

The English is not quite right in multiple places (including some imprecise words such as “vast” and “severity”, unnecessary pluralisation, e.g. “extents”, “infrastructures”) and would benefit from some additional proofreading. Tables and Figures are appropriate (but insert compass roses, and keep format the same among the maps, e.g. scale bars).

Experimental design

The authors highlight that “it is not possible to state the length of a coastline without referring to a specific reference map or to a scale and accuracy of unit of measurement” which is true, but then do not clearly provide this information. We need to know how long the measuring stick is – does OSM provide this? Is it always the same (across countries)? Although OSM seems good for current and future studies, can it be applied to old maps? If not, would it not be better to manually digitise coastlines from old and contemporary maps, do the necessary georeferencing, and then measure distance and areas using exactly the same technique (whatever that may be) on each map?

Validity of the findings

The findings appear to be ok.

Additional comments

Suggest change title to “Expanding walls and shrinking beaches:…”

Line 53. Suggest remove “entire”

Line 61. Not sure soil and concrete are used very much in reclamation, or at least for the actual infill (sand and rock maybe).

Line 63. I thought “land-filling” was burying garbage in big holes? I realise that “land reclamation” as a term has issues, but at least everybody knows what it means… Even if “land-filling” did not describe rubbish dumps, it still wouldn’t be a great descriptor for creating land from the sea (perhaps “sea-filling”?)!

Line 70. I am not sure “nutrients, sedimentation rates, and turbidity” can be degraded.

Line 140. Suggest delete this first sentence.

Line 148. Great that Mandelbrot it cited!

Line 152. Suggest change “are pointless” to “are of little use”

Line 273. Suggest say “over 63%” instead of “considerable part of”

Lines 276-281. This should probably be in the Methods.

Line 286-7. 2x “unique”

Line 310. Perhaps rephrase “extreme proximity to the ocean”

Line 329. Suggest delete “obviously”

Line 340. “Future research should examine how coastal armoring affects the uniqueness of coastal ecosystems.” – I am not sure what this means.

Line 366. Perhaps rephrase “from a coastline-focused point of view”

Line 372. “where the natural coastline has been completely obliterated” is not really the case. Suggest change to “where less than 20% of its natural coastline remains”

375. Perhaps change “like” to “such as”?

378-9. Not sure “easily” or “easy” are quite the right terms. These are big jobs.

385-5. Suggest change “It is our hope that this work will provide a baseline to focus attention on the status quo of the Okinawa Island coastline…” to “It is anticipated this work will focus attention on the current status of the Okinawa Island coastline…”

·

Basic reporting

This is a very good, innovative work. It is a good basis for knowing the current status of the Okinawa Island coast line. I found this could be used for conservation. Also it is a useful information for Okinawan people to think about land use.
Ministry of Environment Japan did their last coastal area survey in 1998 and Okinawa prefecture did similar survey for Okinawan islands from 2009 to 2012, which are quite a while ago.
In both cases the cost of survey had been a problem, but by applying this paper’s way you don’t have to the actual coastal areas and you could obtain data.

Experimental design

no comment

Validity of the findings

1)Line 73 to 90
The author writes about EIA. The author thinks that 'the size of coastal development matters to EIA' . The size of the development is very important factor however it is not only the fact whether to decide EIA should be applied or not. EIA covers only limited laws. Coast Act ('Kaiganhou' in Japanese) is not included in them. Therefore regardless of the size developments for hard armoring could proceed without EIA.

Reference (in Japanese)
https://www.erc-net.com/kankyo/asesu.html#a

'Kouyu suimen umetatehou'(law for landfill in public waters) is included in EIA, so we need EIA for landfills.

2)Line 94 ,95
After EIA process, 'Kouyu suimen umetatehou'(law for landfill in public waters) is applied for landfills. Then about this prefectural governor has power. This is one of the reasons why the governor plays so important for Okinawa.

3) Line 121
The size of US military base at Cape Henoko/Oura Bay has changed from the paper which the author sited. The size of the latest plan for US military base is 160ha.

Reference:
The summary of EIA for Futenma relocation plan
https://www.mod.go.jp/rdb/okinawa/07oshirase/chotatsu/hyoukayouyaku/hyoukayouyaku.html

Additional comments

Orange is applied for Soft armoring and Red is applied for Hard armoring. However these colors are very similar when these two are in one figure. This is a very important thing so I suggest the author to change the color.

---

## Round 0.2 · accepted · Accept

Dear Giovanni and James,

I have carefully reviewed your rebuttal and revised manuscript, and am delighted to accept this interesting study for publication in PeerJ. You will find a number of very small edits in the annotated pdf (attached) that you may incorporate in the proofs.

This study is nicely done and will represent an excellent scientific contribution. Thank you!

With warm regards,
Xavier

Reviewer 2 ·

Basic reporting

Good

Experimental design

Good

Validity of the findings

Good

Additional comments

The revised version looks fine.